# Different NIPBL requirements of cohesin-STAG1 and cohesin-STAG2

Dácil Alonso-Gil[1], Ana Cuadrado[1], Daniel Giménez-Llorente [1], Miriam Rodríguez-Corsino[1] & Ana Losada [1] ✉

Cohesin organizes the genome through the formation of chromatin loops. NIPBL activates cohesin's ATPase and is essential for loop extrusion, but its requirement for cohesin loading is unclear. Here we have examined the effect of reducing NIPBL levels on the behavior of the two cohesin variants carrying STAG1 or STAG2 by combining a flow cytometry assay to measure chromatin-bound cohesin with analyses of its genome-wide distribution and genome contacts. We show that NIPBL depletion results in increased cohesin-STAG1 on chromatin that further accumulates at CTCF positions while cohesin-STAG2 diminishes genome-wide. Our data are consistent with a model in which NIPBL may not be required for chromatin association of cohesin but it is for loop extrusion, which in turn facilitates stabilization of cohesin-STAG2 at CTCF positions after being loaded elsewhere. In contrast, cohesin-STAG1 binds chromatin and becomes stabilized at CTCF sites even under low NIPBL levels, but genome folding is severely impaired.

Cohesin mediates sister chromatid cohesion and organizes the genome through the formation of chromatin loops[1–3]. It consists of four subunits, SMC1A, SMC3, RAD21 and STAG. Additional proteins interact with cohesin and regulate its behavior, most prominently NIPBL, PDS5A/B, WAPL, SORORIN, ESCO1/2, HDAC8 and CTCF. Moreover, two versions of the complex carrying either STAG1 or STAG2 coexist in vertebrate cells and show overlapping and specific functions[4–10]. Importantly, both are required to fulfill embryonic development[11,12]. The two complexes also display different chromatin association dynamics that determine their contributions to loop formation and stability. Cohesin-STAG1 displays longer residence time on chromatin that depends on CTCF and ESCO1 and establishes longer, long-lived chromatin loops together with CTCF[13]. Cohesin-STAG2 shows a preferential interaction with WAPL and mediates shorter loops involved in tissue-specific transcription[7,8,14].

The heterodimer of NIPBL-MAU2 is currently viewed as the cohesin loader, necessary for activation of the cohesin ATPase[15,16]. Cells with low levels of NIPBL have reduced cohesin on chromatin and present altered genome folding, with loss of topological associating domains (TADs) and increased compartmentalization[17–19]. In vitro,

NIPBL is also required both for topological entrapment of plasmid DNA and for loop extrusion by cohesin[20–22]. PDS5 proteins compete with NIPBL for binding cohesin and are, together with WAPL, required for cohesin unloading[23–26]. Thus, NIPBL-bound, presumably loop extruding cohesin, cannot be unloaded. In view of these results, it has been suggested that NIPBL might not function as a cohesin loader but as an extrusion processivity factor that promotes retention of cohesin on chromatin[2]. However, other reports indicate that chromatin remodelers, Mediator or the chromatin regulator BRD4 promote the recruitment and/or stabilization of NIPBL on chromatin, which in turn is important for cohesin loading and function[27–29]. NIPBL is detected preferentially at transcription start sites (TSS) and enhancers, suggesting that these could be loading sites in which nucleosome depletion would facilitate binding of cohesin to DNA[27,30–34].

The two potential functions of NIPBL, loading and extrusion, are difficult to separate. Here, by looking at the specific behavior of cohesin-STAG1 and cohesin-STAG2 after NIPBL knock down (KD), we provide evidence of their different requirements for the putative loader and further speculate that association of cohesin with chromatin may be independent of NIPBL.

[1]Chromosome Dynamics Group, Molecular Oncology Programme, Spanish National Cancer Research Centre (CNIO), Madrid, Spain.
✉e-mail: alosada@cnio.es

## Results

### NIPBL KD does not prevent association of cohesin with chromatin and has opposite effects on STAG1 and STAG2

To assess the presence of cohesin on chromatin throughout the cell cycle in individual cells, we adapted a flow cytometry protocol in which soluble proteins are extracted before fixation and combined it with a barcoding strategy to multiplex samples from different treatments prior to staining[35,36]. As control, we monitored the behavior of the replicative helicase component MCM3, which increases on chromatin during G1 and decreases as S phase progresses while total levels are maintained[37] (Supplementary Fig. 1a, first column). In contrast, the profiles of cohesin subunits were similar in extracted (chromatin-bound) and permeabilized (total) conditions (Supplementary Fig. 1a), consistent with chromatin fractionation results (Supplementary Fig. 1b), and showed massive loading during G1 that further increased during S phase and G2. Strikingly, NIPBL KD only decreased chromatin association of STAG2 while that of STAG1 even increased (Fig. 1a, top, compare colored and gray maps for each protein; quantification in Supplementary Fig. 2a). Immunoblot analysis of chromatin fractions showed some differences although not as clearly and reproducibly as

the flow cytometry (Fig. 1b) while immunostaining further confirmed the opposite behavior of the two cohesin variants after NIPBL KD (Fig. 1c and Supplementary Table 1). Similar results were obtained in two additional cell lines (Supplementary Fig. 2b, c) and with different siRNAs (Supplementary Fig. 2d). To exclude a cohesin-independent role of STAG1, we co-depleted SMC1A along with NIPBL. The increase of STAG1 on chromatin was abrogated under this condition, suggesting that it occurs in the context of a full cohesin complex (Fig. 1d). Taken together, our results clearly show that a strong reduction in NIPBL levels does not prevent the association of cohesin with chromatin but affects the two cohesin variants in opposite ways.

### Different chromatin association dynamics of STAG1 and STAG2 do not dictate their different response to NIPBL KD

We reasoned that the increased presence of STAG1 on chromatin in the NIPBL KD condition could be the result of its more stable association, as a more stable complex would be less dependent on the loader. However, increased chromatin association of cohesin-STAG1 persisted after co-depletion of NIPBL and either CTCF or ESCO1 (Fig. 2a and Supplementary Fig. 3). Similarly, reducing the dynamic behavior of

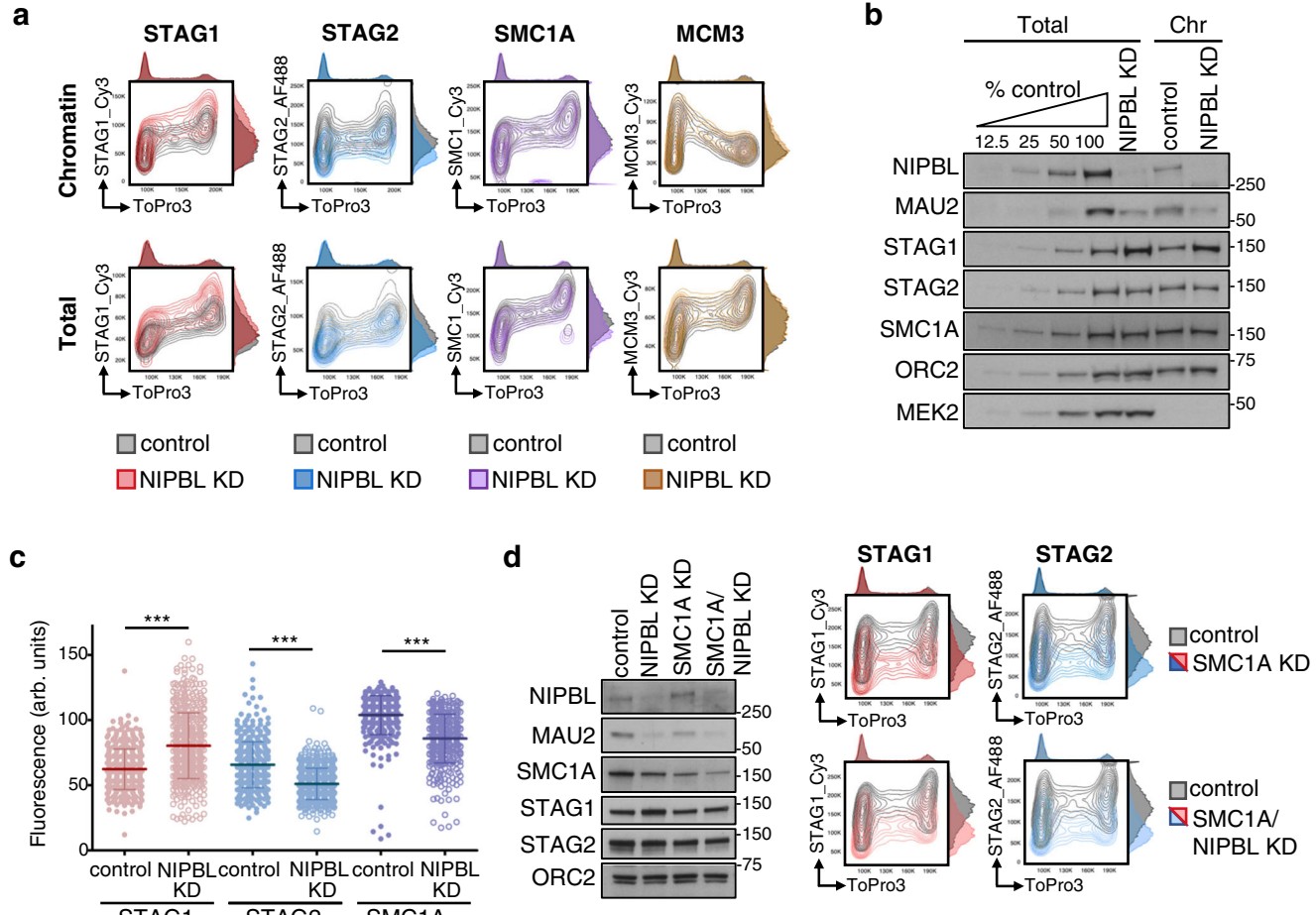

**Fig. 1 | NIPBL KD affects cohesin-STAG1 and cohesin-STAG2 in opposite ways.**
**a** Asynchronously growing HeLa cells mock transfected (control) or transfected with siRNA against NIPBL (NIPBL KD) were analyzed by flow cytometry 72 h later. Contour plots of the indicated proteins in control (gray plots) and NIPBL KD cells (colored plots) were overlapped for comparison. For each map, the cell cycle profile according to DNA content appears on top while the distribution of antibody intensities is plotted on the right. **b** Immunoblot analysis of chromatin fractions (Chr) and total cell extracts from control and NIPBL KD cells. Increasing amounts of total extract from control cells were loaded to better quantitate the extent of depletion. NIPBL partner MAU2 also decreases after NIPBL KD. This is one

representative experiment of at least 3 performed. **c** Quantitative immuno-fluorescence (arb. units, arbitrary units) of control or NIPBL KD HeLa cells stained with antibodies against STAG1, STAG2 and SMC1A. At least 372 cells were analyzed per condition in a single experiment. Means and SD are plotted. A non-parametric Mann–Whitney two-sided test with confidence intervals of 99% was performed. ***$p < 2e-16$. See also Supplementary Table 1. **d** Flow cytometry contour plots for chromatin-bound STAG1 and STAG2 in control (gray contour plots), SMC1 KD and double NIPBL/SMC1A KD (colored contour plots) HeLa cells. The immunoblot on the left shows remaining protein levels in total cell extracts in the different conditions. The experiment was performed twice with similar results.

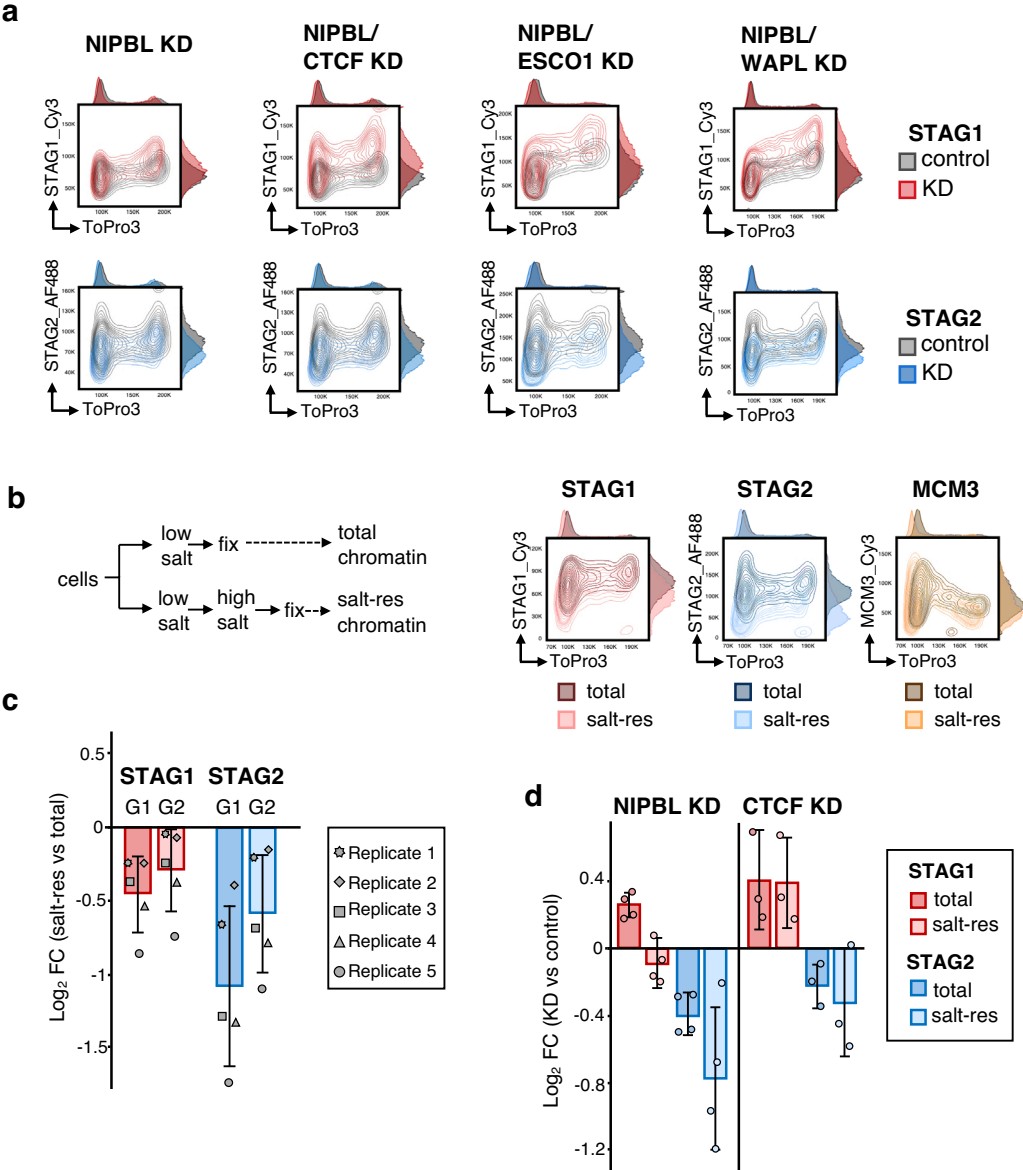

**Fig. 2 | Effect of cohesin regulators on the response of cohesin variants to NIPBL KD. a** Representative flow cytometry contour plots for chromatin-bound STAG1 and STAG2 in control (gray plots) and KD (colored plots) HeLa cells. For each double KD condition, a single KD condition was also analyzed (see Supplementary Fig. 3). The NIPBL KD plot shown corresponds to the experiment co-depleting NIPBL and CTCF. Experiments were performed three times with similar results. **b** Left, scheme of the experiment. Right, flow cytometry contour plots comparing total and salt-resistant chromatin-bound levels of STAG1, STAG2 and MCM3.

**c** Quantification of salt-resistant vs. total chromatin-bound levels of STAG1 and STAG2 in G1 and G2 (*n* = 5 experiments). The graph shows mean and standard deviation (SD) of the log2 fold change (log2FC) of median antibody intensity (salt res vs. total). **d** Changes in total and salt-resistant chromatin-bound levels of STAG1 and STAG2 in G1 cells after NIPBL KD (*n* = 4 experiments) or CTCF KD (*n* = 3 experiments). The graph shows mean and SD of the log2FC of median antibody intensity (KD vs. control).

cohesin-STAG2 by co-depleting WAPL together with NIPBL did not alter the chromatin flow cytometry profiles of either variant compared with the single depletion of NIPBL (Fig. 2a and Supplementary Fig. 3). Thus, the opposite effect of NIPBL KD on chromosome-bound levels of cohesin-STAG1 and cohesin-STAG2 is not simply a consequence of their different chromatin association/dissociation dynamics imposed by CTCF, ESCO1 and WAPL.

To assess the "quality" of the chromatin association measured by flow cytometry, we modified the protocol to include an incubation in high-salt buffer before fixation. This extra step reduced the amounts of both STAG1 and STAG2 on chromatin, but affected STAG2 more severely, further supporting their different behavior. In contrast, the MCM3 profile was unchanged (Fig. 2b). When we

segregated G1 and G2 cells, we observed that the difference between the salt resistant fraction and the total chromatin-bound protein was reduced in G2 for both STAG1 and STAG2, consistent with the stabilization resulting from cohesion establishment (Fig. 2c). We focused then on G1 to avoid interference of this cohesive cohesin. After NIPBL KD, the increase in "total" chromatin-bound STAG1 was not paralleled by an increase in "salt-resistant" STAG1 (Fig. 2d, left). For STAG2, there was little protein left in NIPBL KD cells after the high-salt incubation. For comparison, we repeated the experiment with CTCF KD cells, a condition that also increases the amount of STAG1 on chromatin (Supplementary Fig. 3c). In this case, however, "salt-resistant" and "total" chromatin-bound STAG1 increased to the same extent (Fig. 2d, right). We conclude that NIPBL is required to

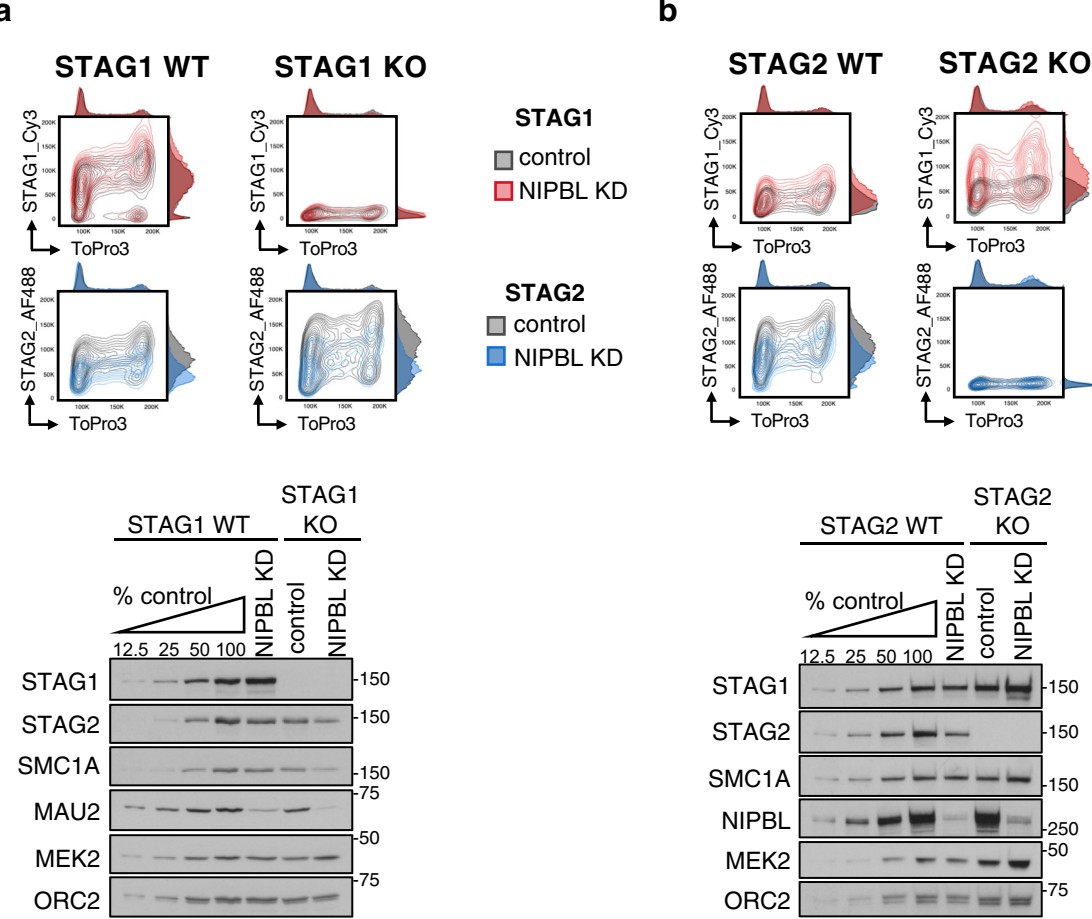

**Fig. 3 | NIPBL KD increases cohesin on chromatin in STAG2 KO cells. a** Top, contour plot profiles of chromatin-bound STAG1 and STAG2 in A673 cells with (WT) or without (KO) STAG1 in control (gray) and NIPBL KD (colored) condition. Bottom, immunoblot analyses of the same cells. **b** As in **a**, for A673 cells with (WT) or without (KO) STAG2. All experiments were performed three times with similar results.

stabilize the binding of both cohesin variants to chromatin, maybe by promoting topological entrapment, even before cohesion establishment. Recent results suggest that (pseudo) topological embrace may be important for CTCF-cohesin loops[38].

**The response of cohesin-STAG1 and cohesin-STAG2 to NIPBL KD does not depend on the presence of the other variant**

We next asked about the crosstalk between STAG1 and STAG2 loading. In the presence of NIPBL, the reduction of one of the variants did not affect significantly the amount of the other variant on chromatin (Supplementary Fig. 4). We also generated A673 cells carrying a single complex and found that when only cohesin-STAG2 is present (STAG1 KO cells), NIPBL KD still reduced cohesin levels on chromatin (Fig. 3a). More importantly, in cells with only cohesin-STAG1 (STAG2 KO cells), reduction of the putative cohesin loader increased the amount of chromatin-bound cohesin (Fig. 3b). We conclude that NIPBL promotes or stabilizes the association of STAG2 with chromatin but restricts that of STAG1. The latter effect is independent of the presence of STAG2 and may rely, at least in part, on repression of *STAG1* transcription, as increased *STAG1* mRNA levels are detected after reduction of NIPBL in several contexts, including blood cells from Cornelia de Lange (CdLS) patients with *NIPBL* mutations[18,19,39–41] (Supplementary Table 2). A corollary from the results presented so far is that at least cohesin-STAG1 complexes have the ability to associate with chromatin independently of NIPBL, either on its own or aided by a different loader yet to be identified.

**Cohesin-STAG1 further accumulates at CTCF sites upon reduction of NIPBL levels**

Current models propose that cohesin is loaded at sites bound by NIPBL, often TSS or active enhancers, and then moves away extruding DNA until stopped and stabilized at CTCF sites[18,31,34]. Given the requirement of NIPBL for loop extrusion, we expected that cohesin-STAG1 complexes present on chromatin in NIPBL KD cells would be less able to reach CTCF sites and would instead accumulate at their loading sites. Calibrated chromatin immunoprecipitation sequencing (ChIP-seq) in MCF10A cells showed that STAG1 was present, and even increased, at the same CTCF-bound sites in NIPBL KD and control cells. In contrast, STAG2 signals were significantly decreased at both CTCF and non-CTCF cohesin positions (Fig. 4a, c). As a result of this opposite behavior of the two variants, total cohesin detected with anti-SMC1A was reduced genome-wide, but not to the same extent as STAG2 (Supplementary Table 3). Assuming that NIPBL is required for loop extrusion, the most likely explanation for this result is that cohesin-STAG1 binds chromatin preferentially at or near CTCF positions. Strikingly, after CTCF KD to around 20% of its normal levels, cohesin-STAG1 still accumulates at CTCF sites in which remaining CTCF is also bound while STAG2 is drastically reduced, similar to what happens in NIPBL KD condition (Fig. 4b, c and Supplementary Fig. 5a). We speculate that cohesin-STAG1 associates with chromatin in a NIPBL-independent manner at CTCF sites and its interaction with CTCF prevents WAPL-mediated unloading. In contrast, a significant fraction of cohesin-STAG2 may be loaded elsewhere in the genome, possibly also without NIPBL, but requires NIPBL to arrive to CTCF

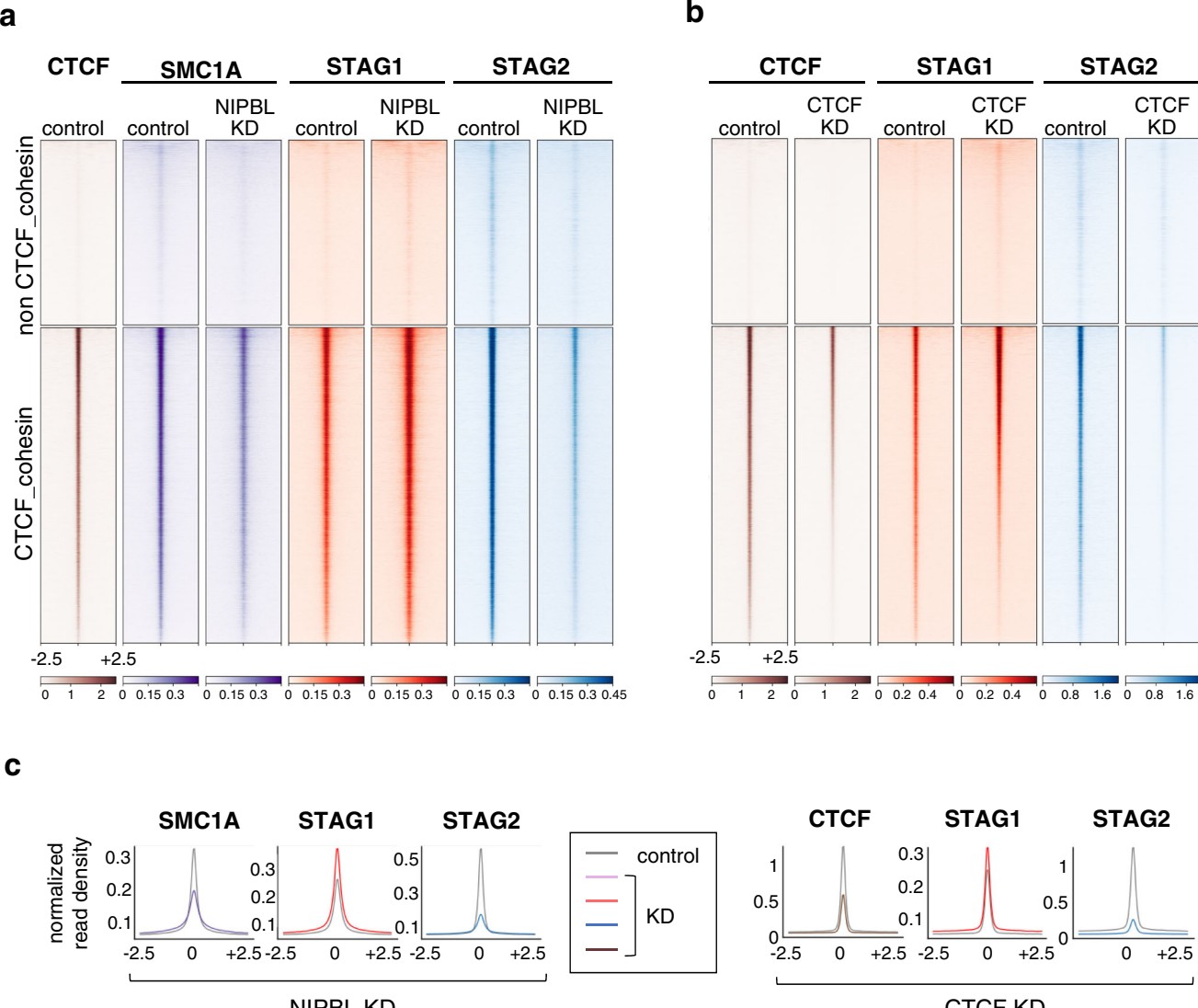

**Fig. 4 | Cohesin-STAG1 accumulates at CTCF sites in NIPBL KD cells.**
**a** Distribution of three cohesin subunits in control and NIPBL KD MCF10A cells. Reads from calibrated ChIP-seq are plotted in a 5-kb window centered in the summits of cohesin positions with and without CTCF (24,912 and 14,607 positions, respectively). Information on replicates and additional datasets used in Supplementary Table 4. **b** Distribution of CTCF, STAG1 and STAG2 in control and CTCF KD MCF10A cells, as in **a**. For a replicate experiment and additional analyses see Supplementary Fig. 5. **c** Normalized read density plots for cohesin subunits ±2.5 kb of the summit in the different KD conditions from the heatmaps above.

sites by loop extrusion. In cells with full NIPBL but reduced CTCF levels, *STAG1* is upregulated (Supplementary Fig. 3b) and cohesin-STAG1 preferentially occupies remaining CTCF-bound positions (Supplementary Fig. 5b). Even if NIPBL-bound cohesin-STAG2 can reach those sites, they would be already occupied by cohesin-STAG1 and STAG2 would not arrest there[5]. It is also possible that the STAG2-CTCF interaction is outcompeted by WAPL binding more easily than the STAG1-CTCF interaction, particularly when the levels of CTCF are reduced, consistent with our previous proposal of differential affinities of STAG1 and STAG2 for CTCF and WAPL[8]. Co-depletion of NIPBL and CTCF further reduced the presence of cohesin-STAG2 genome-wide while cohesin-STAG1 was maintained at CTCF sites (Supplementary Fig. 5c–e).

### Cohesin-STAG1 cannot form loops in the absence of NIPBL
We next asked if cohesin-STAG1 present at CTCF sites in NIPBL KD cells was able to form and extrude loops. For that, we performed in situ Hi-C analyses in mock transfected (control) MCF10A cells, and cells treated with different siRNAs against NIPBL (Supplementary

Fig. 6a, b). After confirmation of a high correlation among replicates, data from control (3 replicates) and NIPBL KD cells (4 replicates) were merged for subsequent analyses (Supplementary Fig. 6c and Supplementary Table 5). Interaction frequencies in the 0.1–1.2 Mb range decreased in NIPBL KD cells compared to control cells and increased at higher genomic distances, suggesting loss of cohesin-mediated loops and enhanced compartmentalization (Fig. 5a, left and Supplementary Fig. 6d). The latter was confirmed by visual inspection of Hi-C matrices of whole chromosomes in which the checkerboard pattern was better defined in the NIPBL KD condition (Fig. 5a, I). Zooming in, many loops seen in control cells were reduced or lost in NIPBL KD cells (Fig. 5a, II). Genome-wide, the number of called loops decreased after KD, and among differential loops detected in the two conditions, lost loops were clearly longer than shared and gained loops (Fig. 5b and Supplementary Data 1). They also showed a smaller increase in STAG1 occupancy at their anchors and, consequently, a larger loss in total cohesin (Supplementary Fig. 6e). Metaplots of loops of different sizes confirmed some gain of interactions at short distances, very close to loop

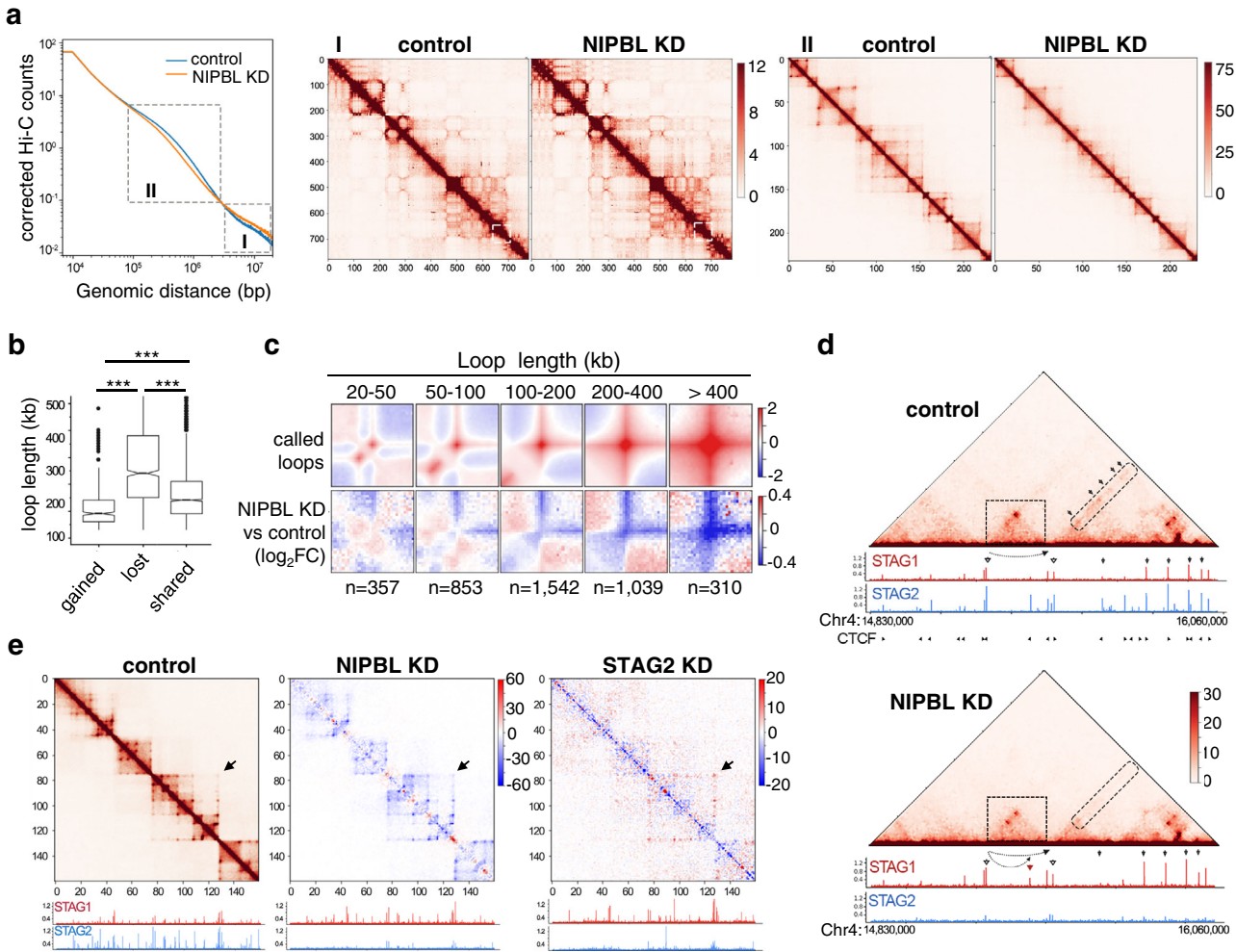

**Fig. 5 | NIPBL is required for loop extrusion. a** Contact probability as a function of genome distance in control and NIPBL KD cells (left) and normalized contact matrices for whole chromosome 17 (I) and the boxed region within (II; Chr17:66,700,000–72,500,000) that exemplify changes at very long (I) and TAD-scale (II) distances. Resolution is 100 kb/bin in I and 25 kb/bin in II. In situ Hi-C data come from three replicates in mock transfected (control) MCF10A cells and four replicates in NIPBL KD cells (see Supplementary Fig. 6). **b** Box plot for the size of gained (406), lost (1029) and shared (2666) loops called at 10-kb resolution between control and NIPBL KD cells (see "Methods" and Supplementary Data 1 for genomic coordinates). Boxes represent interquartile range (IQR); the midline represents the median; whiskers are 1.5 × IQR; and individual points are outliers. A non-parametric Mann–Whitney two-sided test and Holm's correction for multiple comparisons was used. ***$p < 2e−16$. **c** Metaplots for loops of the indicated sizes in

control cells (top) and how they change after NIPBL KD (bottom). The number of loops in each category is indicated below. **d** Representative region in chromosome 4 (chr4:14,830,000–16,060,000) showing contacts (10-kb resolution), distribution of STAG1 and STAG2, and CTCF positions and orientation (top matrix only). In the center of the matrix (boxed) a long loop decreases and a shorter one within slightly increases in the NIPBL KD condition (dashed arrows below the matrix). On the right, a stripe that is reduced indicated. Arrowheads signal cohesin positions. **e** Comparison of the differential contacts observed in NIPBL KD and STAG2 KD cells (25 kb/bin) in a region of chromosome 7 (chr7:24,000,000–28,000,000) and corresponding distribution of STAG1 and STAG2, as measured by ChIP-seq. The matrix shown on the left corresponds to the control of NIPBL KD cells. Arrow points to a 1.3 Mb-long loop that disappears in NIPBL KD but is maintained (even increased) in STAG2 KD cells[8].

anchors, while for longer loops interactions were drastically reduced in NIPBL KD cells (Fig. 5c). We reckon that remaining NIPBL levels in these cells may allow cohesin-STAG1 to perform loop extrusion to certain extent, forming short loops, but further extension is severely impaired. An example is shown in Fig. 5d, in which a prominent loop in the center of the matrix in control cells is clearly reduced while a shorter loop is maintained and even slightly increased after NIPBL KD. Moreover, next to this loop one can observe the disappearance of a stripe after NIPBL KD (Fig. 5d). These changes in chromatin organization are not just the result of reduced levels of cohesin on chromatin (Supplementary Fig. 6b). Changes in STAG1 and STAG2 genome-wide distribution after NIPBL KD are similar to those previously observed in STAG2 KD cells, with little accumulation of STAG2 anywhere in the genome and STAG1 present mainly at CTCF-cohesin sites and slightly increased with respect to the control

condition[8] (Fig. 5e, snapshots of the genome browser appear below matrices). Despite this similar distribution of cohesin, differential Hi-C matrices provide evidence for the specific changes in chromatin folding caused by NIPBL KD and STAG2 KD. In particular, we observed that longer loops, which depend specifically on STAG1 and are prominent in STAG2 KD cells[8,13] do require also NIPBL since they are lost in NIPBL KD cells (Fig. 5e). Consistent with more dramatic changes in chromatin architecture in NIPBL KD cells, three times more differentially expressed genes (DEGs) were detected in these cells, although a number of them were common between the two conditions [3340 DEGs in NIPBL KD; 1154 in STAG2 KD; 685 common DEGs; Supplementary Fig. 7a, b and Supplementary Data 2 and 3]. Importantly, the transcriptional changes observed after NIPBL KD, but not after STAG2 KD, resemble those found in blood cells from CdLS patients carrying *NIPBL* mutations[40] (Supplementary Fig. 7c).

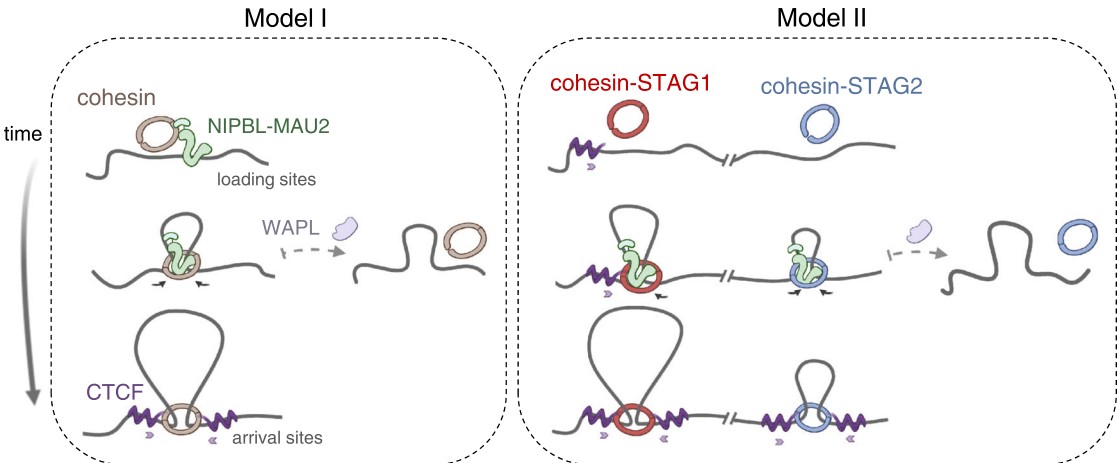

**Fig. 6 | An alternative model for the role of NIPBL in chromatin association and extrusion by the two cohesin variants.** In model I, NIPBL-MAU2 promotes chromatin association of cohesin (STAG1 or STAG2) as well as loop extrusion until the complex is released by WAPL or becomes arrested by CTCF proteins bound in convergent orientation. In the alternative model II, NIPBL-MAU2 is not required for association of cohesin with chromatin, but it is for loop extrusion. Cohesin-STAG1 binds at/near CTCF sites while cohesin-STAG2 is loaded elsewhere and requires NIPBL to reach them. Image created with BioRender.com.

## Discussion

Here we have addressed changes in chromatin association and genome-wide distribution of cohesin-STAG1 and cohesin-STAG2 after NIPBL KD in human cells. We have found that cohesin-STAG1 levels increase in this condition and the complex further accumulates at CTCF sites although it cannot form long loops. In contrast, cohesin-STAG2 levels decrease genome-wide. These opposite effects on the two variants are independent of the presence of the other variant and epistatic to KD of other regulators of cohesin dynamics. Previous results in yeast had shown that cohesin could be detected at loading sites in *scc2* mutants by chromatin immunoprecipitation[42]. Also, downregulation of MAU2 by siRNA in HeLa cells or its complete knock out (KO) in HAP1 cells reduced considerably the amount of NIPBL but left significant amounts of cohesin on chromatin[17,18]. Even genetic deletion of *NIPBL* in mouse liver cells led to a more severe reduction of SMC1 on chromatin than of STAG1[19]. Thus, our results are consistent with previous data showing that significant amounts of cohesin can still be found on chromatin after knock down of NIPBL or MAU2.

While we cannot discard that the small amount of NIPBL left after KD may be sufficient to load all the cohesin that we detect bound to chromatin under this condition, an alternative possibility is that association of cohesin with chromatin does not require NIPBL (Fig. 6). Instead, binding of NIPBL to cohesin promotes retention of the complex on chromatin, an effect that is particularly important for cohesin-STAG2. To date, it is unclear how cohesin engages with DNA to perform its different functions in 3D genome organization and cohesion[38,43–48]. Even entrapment of DNA by cohesin can take place in the absence of NIPBL, at least in vitro[20,49]. Structural studies suggest that cohesin has two DNA binding modules, the STAG/hinge module and the NIPBL/SMC head module and it is possible that the former is sufficient for transient chromatin association although not for functional translocation of the complex[46,47,50]. We propose that this initial and less stable association of cohesin with DNA, independent of NIPBL, can be detected in our flow cytometry assay. When cells are challenged with extra salt, a decrease in chromatin-bound cohesin is observed also for cohesin-STAG1 in NIPBL KD cells, although not as pronounced as the decrease in cohesin-STAG2. Importantly, SMC1A KD decreases the binding of both variants to chromatin without changing their cellular levels, which supports that the assay detects bona-fide chromatin association of the whole complex and not unspecific association of the cohesin subunits (Fig. 1d). The decrease in SMC1A accumulation at

CTCF sites observed in NIPBL KD cells by ChIP-seq, much milder than the decrease in STAG2, further suggests that the whole cohesin-STAG1 complex, and not only STAG1, is present at these sites under this condition, although re-ChIP experiments would be required to confirm this point (Fig. 4c).

Upon arrival to CTCF sites, cohesin becomes resistant to WAPL-mediated unloading[51]. We postulate that cohesin-STAG1 associates with chromatin near CTCF sites or arrives there and becomes arrested even when NIPBL levels are very low (Fig. 6). This fact, together with the transcriptional upregulation observed after NIPBL KD, can explain the increased presence of this complex on chromatin. Unlike cohesin-STAG1, cohesin-STAG2 may be preferentially loaded at sites devoid of CTCF, and thus requires binding to NIPBL to translocate and reach CTCF positions. Faster arrival of cohesin-STAG1 complexes to CTCF sites is also consistent with the prevalence of STAG1 over STAG2 at those CTCF-bound sites remaining after CTCF KD (Fig. 4b and Supplementary Fig. 5). Whether the longer residence time of cohesin-STAG1 on chromatin is sufficient to explain our results is unclear. This residence time depends on ESCO1 and CTCF but we here show that KD of any of these two factors along with NIPBL does not modify the results of single NIPBL KD. Likewise, WAPL KD to the levels achieved here (around 25% of normal levels) cannot reverse the NIPBL KD effect on cohesin-STAG2. This is in contrast to results in HAP1 cells that show how WAPL KO can rescue the effects of low NIPBL levels in MAU2 KO cells[18]. It is likely that the relative amounts of the two cohesin variants as well as cohesin regulators in different cells and conditions (KD vs. KO) affect the final outcome of perturbation experiments. We would like to emphasize, however, that the opposite response of STAG1 and STAG2 to NIPBL KD is similar in the three human cell lines tested here. Importantly, our results confirm the importance of NIPBL for genome folding in vivo[18,19], as cohesin-STAG1 complexes found at CTCF positions in NIPBL KD cells have reduced ability to engage in chromatin loop formation.

## Methods
### Cell culture
HeLa and A673 cells were cultured in DMEM (BE12-604F/U1, Lonza) supplemented with 10% FBS and 1% penicillin-streptomycin. MCF10A cells were cultured in DMEM/F12 (#31330038, Thermo Fisher) supplemented with 20 ng/ml EGF, 0.5 mg/ml hydrocortisone, 100 ng/ml cholera toxin, 10 µg/ml insulin and 5% horse serum. All cell lines were grown at 37 °C under 90% humidity and 5% $CO_2$.

### siRNA treatment

HeLa, A673 and MCF10A cells were transfected with 50 nM siRNAs (Supplementary Table 6) using DharmaFECT reagent 1 and Gibco Opti-MEM I Reduced Serum Media (#31985047 Thermo Fisher). Cells were harvested 72 h after transfection and analyzed by flow cytometry, as described below. Protein and mRNA levels were assessed by immunoblotting and quantitative RT-PCR, respectively.

### CRISPR-Cas9 editing

A673 cells expressing inducible Cas9 (A673_iCas9) were generated as described[52]. A single cassette containing both the rTetR activator under CAG promoter and the Tetracycline Response Element (TRE) promoter driving the expression of Cas9, was inserted in the *AAVS1* locus by homologous recombination using the Cas9 nuclease and a guide RNA sequence (gRNA 5′-GGGGCCACTAGGGACAGGAT-3′) against intron 1 of *AAVS1*. To generate STAG1 and STAG2 KO cell lines, viruses were produced through transfection of $3 \times 10^6$ 293T cells with 9 μg of lentiGuide-Puro (Addgene, #52963) containing guides for STAG1 or STAG2 (Supplementary Table 6), 5 μg of psPAX2 and 2.5 μg of pMD2G in Gibco Opti-MEM I Reduced Serum Media with Lipofectamine 2000 Transfection Reagent. After 48 h, viruses were purified through centrifugation at $600 \times g$ and filtered. A673_iCas9 cells (considered WT cells for the experiments in Fig. 3) were grown in medium with 2 μM doxycycline to induce Cas9 expression and 8 μg/ml of polybrene and transduced with viruses. After 72 h, cells were seeded at low density for clonal selection. Clones were analyzed by immunoblotting with STAG1 and STAG2 antibodies to check protein elimination.

### Antibody generation

A rabbit polyclonal antibody was produced against the C-terminal region of mouse NIPBL (aa 2349-2667) and affinity purified. Rat monoclonal antibodies were raised against a recombinant fragment of human CTCF (aa 574-727) and human WAPL (aa 838-1190). These and additional antibodies used in this study are listed in Supplementary Table 7.

### Western blotting and chromatin fractionation

Cells were collected by trypsinization, counted and resuspended in RIPA buffer at $10^7$ cells/ml for 30 min. Upon centrifugation at $14,000 \times g$, supernatant was taken, SDS-loading buffer was added and samples boiled. Equal volumes were separated by SDS-PAGE in NuPAGE™ 3–8% Tris-Acetate gels (#EA0375PK2). Alternatively, samples were resuspended in SDS–PAGE loading buffer at $10^7$ cells/ml, sonicated and boiled before fractionation in a 7.5% SDS-PAGE gel. Gels were transferred to nitrocellulose membranes in Transfer buffer I (50 mM Tris, 380 nM Glycine, 0.1% SDS, 20% methanol) for 1 h at 100 V and analyzed by immunoblotting. Antibodies and dilutions are listed in Supplementary Table 7. Chromatin fractionation was performed as described[37]. Uncropped and unprocessed scans of all blots are shown in the Source Data file.

### Immunofluorescence

Cells grown on coverslips coated with poly-Lysine were pre-extracted in CSK buffer (10 mM PIPES pH 7, 0.1 M NaCl, 0.3 M sucrose, 3 mM MgCl₂, 0.5 mM PMSF) with 0.5% Triton X-100 for 5 min before fixation for 15 min in a 2% formaldehyde solution. After incubation for 5 min in CSK-0.5% TX-100, coverslips were blocked with 3% BSA-0.05% Tween-20 in PBS for 30 min. Primary and secondary antibodies were diluted in blocking solution and incubated for 1 h each at room temperature (RT). DNA was counterstained with 1 μg/ml DAPI. Images were acquired in a TCS-SP5 (AOBS) Confocal microscope (Leica Microsystems) with LAS AF v2.6 acquisition software. Images were analyzed with a custom made software programed in Definiens Developer XD v2.5 software (Definiens).

### Flow cytometry assay

Flow cytometry assays were performed as described[35] with some modifications. To analyze chromatin bound proteins, cells were treated for 5 min with a low salt extraction buffer (0.1% Igepal CA-630, 10 mM NaCl, 5 mM MgCl₂, 0.1 mM PMSF, 10 mM Potassium Phosphate buffer pH 7.4) and fixed in 1% PFA final concentration. To evaluate the strength of chromatin association, cells were incubated for 5 min with salt extraction buffer containing 100 mM NaCl after the 5 min in low salt extraction buffer and before fixation. To analyze total proteins, unextracted cells were fixed in ice-cold 70% ethanol for 2 h. To eliminate antibody staining variation among samples from different conditions, a barcoding strategy was used[36]. Four different samples were stained with increasing dilutions of Pacific Blue (Invitrogen) for 30 min in the dark at RT and then mixed into one tube. Then, each barcoded sample was blocked in flow buffer (0.1% Igepal CA-630, 6.5 mM Na₂HPO₄, 1.5 mM KH₂PO₄, 2.7 mM KCl, 137 mM NaCl, 0.5 EDTA pH 7.5, 4% non-fat milk) for 5 min and consecutively incubated with primary and secondary antibodies, also diluted in flow buffer, for 1 h each. Finally, DNA staining was performed over night with 125 nM ToPRO3-iodide 642/661 in PBS.

Cells were analyzed on a BD LSRII Fortessa flow cytometer using BD FACSDiva software and four different lasers: *680/30_R* laser for ToPRO3 (DNA), *450/50_V* for Pacific Blue (barcoding), *586/15_YG* for Cy3-labeled secondary antibody and *525/50_B* laser for Alexa fluor 488-labeled secondary antibody. For statistical analysis, single cell cycles were gated and at least 10,000 cells were recorded for each population in a barcoded sample (Supplementary Fig. 8). For imaging data, the same number of events were exported for each barcoded population in a FlowJo v10 software. Data quality and fluorescence compensation were assessed in order to correct for emission spectra overlap.

### Quantitative RT-PCR

cDNAs were generated using the Superscript II Reverse Transcriptase (Invitrogen) from total RNA (RNeasy Mini Kit, Qiagen) and qRT-PCR analyses were performed using the SYBR Green PCR Master Mix and an ABI Prism® 7900HT instrument (Applied Biosystems®). Reactions were performed in triplicate for each sample and samples came for at least three experiments. Expression was normalized to that of the endogenous housekeeping gene *GAPDH*, using the ΔΔCt method. Primers used are listed in Supplementary Table 6.

### Chromatin-Immunoprecipitation assay

MCF10A cells were grown at high confluence in order to arrest them in G1. Cells in suspension were crosslinked with 1% formaldehyde added to the media for 15 min at RT. After quenching the reaction with 0.125 M Glycine, fixed cells were washed twice with PBS containing 1 μM PMSF and protease inhibitors. For chromatin preparation two different protocols were applied depending on the experiment. For experiments labeled in blue in Supplementary Table 4, cells were pelleted and lysed in lysis buffer (1% SDS, 10 mM EDTA, 50 mM Tris-HCl pH 8.1) at a concentration of $2 \times 10^7$ cells/ml. Sonication was performed with a Covaris S220 (shearing time 30 min, 20% duty cycle, intensity 6, 200 cycles per burst and 30 s per cycle) in a minimum volume of 2 ml. For other experiments, nuclei were isolated before sonication. Cells were incubated for 10 min at 4 °C in 25 ml ice-cold buffer A (10 mM HEPES pH 8.0, 10 mM EDTA pH 8.0, 0.5 mM EGTA, 0.25% Triton X-100 and protease inhibitors), recovered by centrifugation, resuspended in ice-cold buffer B (10 mM HEPES pH 8.0, 200 mM NaCl, 1 mM EDTA, 0.5 mM EGTA, 0.01% Triton X-100 and protease inhibitors) for 10 min at 4 °C and centrifuged again. Nuclei were lysed in chromatin lysis buffer (50 mM Tris-HCl pH 8.0, 10 mM EDTA, 0.25% SDS and protease inhibitors) at a concentration of $2 \times 10^7$ nuclei/ml and stored overnight at 4 °C. Sonication was performed in a Covaris E220 device (shearing time 7 min at 5–7 °C range with 140 peak incident power, 5% duty factor, 200 cycles per burst). In all cases, chromatin from $10^7$ cells was

used per immunoprecipitation reaction with 50 μg of cohesin antibodies or 15 μg of CTCF antibody as described[8]. For calibration, 5% of chromatin from mouse ES cells was added to the human chromatin. For library preparation, at least 5 ng of DNA were processed through subsequent enzymatic treatments with "NEBNext Ultra II FS DNA Library Prep Kit for Illumina" from New England BioLabs (cat# E7805). Briefly, a short fragmentation of 10 min was followed by end-repair, dA-tailing, and ligation to adapters. Adapter-ligated libraries were completed by limited-cycle PCR (8–12 cycles). Resulting average fragment size is 300 bp from which 120 bp correspond to adapter sequences. Libraries were applied to an Illumina flow cell for cluster generation and sequenced on Illumina NextSeq 500 (with v2.5 reagent kits) following manufacturer's recommendations.

### ChIP-sequencing analysis

Alignment of reads to the reference human genome (hg19) was performed using "Bowtie2" (version 2.4.2) under default settings[53]. Duplicates were removed using GATK4 (version 4.1.9.0). Reads were then plotted around "CTCF" and "non-CTCF" cohesin positions defined from our previously published data for STAG1, STAG2 and SMC1A[8] and CTCF peaks[54] as follows. First, peaks were called for each protein with MACS2 (version 2.2.7.1) after setting the $q$ value (FDR) to 0.05 and using the "–extsize" argument with the values obtained in the "macs2 predictd" step[55]. Then, cohesin peaks called for any of the three subunits were merged and intersected with CTCF peaks to define two clusters of cohesin positions with or without CTCF. For analysis of calibrated ChIP-seq, profiles for each antibody were normalized by coverage and then multiplied by the occupancy ratio $(OR) = (W_m IP_h)/(W_h IP_m)$, where $W_h$ and $IP_h$ are the number of reads mapped to the mouse genome from input (W) and immunoprecipitated (IP) fractions, and $W_m$ and $IP_m$ are reads mapped to the human genome from the input and IP fractions used for calibrating[56]. Mean read-density profiles and read-density heatmaps for different chromatin-binding proteins were generated with deepTools 3.5.0[57].

### In situ Hi-C

MCF10A cells ($3 \times 10^6$ cells per condition) were fixed with 2% formaldehyde (Sigma, #252549) in PBS. Formaldehyde was quenched with 300 mM of glycine at RT for 5 min. Hi-C experiments were performed with Arima-HiC Kit (#A510008) following manufacturer's instructions. Libraries were prepared with Swift Biosciences Accel-NGS 2S Library Kit (Cat# 21024) and amplified with KAPA Library Amplification Kit (KAPA Cat# KK2620). For all the libraries, 8–9 PCR cycles were used for amplification and they were then sequenced on an Illumina NextSeq550 (82 × 43 bp).

### Hi-C analysis

Sequences were aligned to the reference human genome (hg19) using "Bowtie2" with–local–reorder flags. Then, 5-kb raw matrices were built using hicBuildMatrix from HiCExplorer[58]. To assess the reproducibility of the replicates, 5-kb raw matrices were summed to obtain 40-kb matrices and these were normalized first by coverage and then by KR algorithm. Given the high correlation of the distribution of contacts as a function of genomic distance among replicates for each condition (control or KD), replicates were merged and analyzed at 5-kb resolution. Contacts at higher resolution (10-kb, 20-kb, 100-kb, etc.) were obtained by summing contacts at lower resolution followed by normalization by coverage and KR.

Loops were called at 10-kb resolution in each replicate and only those loops that were called at least twice among all the replicates were considered for subsequent analyses. We next defined "gained" loops as those called only in NIPBL KD replicates and "lost" loops as those called only in wild type replicates. The rest of the loops were considered "shared" loops. Genomic coordinates for all these loops can be found in Supplementary Data 1. For the boxplot analysis in Supplementary

Fig. 6e, signals for each protein and condition were obtained ±150-bp around peak summits and log2 fold change between NIPBL KD and control was calculated. Next, we intersected these peaks with anchors of gained, lost and shared loops. The metaplots of the loops in Fig. 5c were obtained using coolpup.py[59].

### Bulk RNA sequencing

Total RNA was extracted with NZY Total RNA Isolation kit (MB13402) following manufacturer's instructions. Total RNA samples (500 ng) were processed with the "NEBNext Single Cell/Low Input RNA Library Prep" kit (NEB #E6420) by following manufacturer instructions. RNA Quality scores were 9.9 on average (range 9.1–10) when assayed on a PerkinElmer LabChip analyzer. Briefly, an oligo(dT) primed reverse transcription with a template switching reaction was followed by double stranded cDNA production by limited-cycle PCR. Non-directional sequencing libraries were completed with the "NEBNext Ultra II FS DNA Library Prep Kit for Illumina" (NEB #E7805) and subsequently analyzed on an Illumina NextSeq 550 with v2.5 reagent kits following manufacturer's protocols.

### RNA-sequencing analysis

Fastq files with 86-nt single-end sequenced reads were quality-checked with FastQC (S. Andrews, http://www.bioinformatics.babraham.ac.uk/projects/fastqc/) and aligned to the human genome (hg19) with Nextpresso[60] executing TopHat-2.0.0 using Bowtie 0.12.7 and Samtools 0.1.16 allowing two mismatches and five multi-hits. The reads were mapped to hg19 genes using HTSeq and the differential expression was obtained using the R package DESeq2[61]. We consider that a gene is expressed if the mean of the reads for the replicates is greater than 2 and changes in the expression of those genes are significant if FDR < 0.05 and absolute log2 fold change >0.5. The same analysis was applied to data from NIPBL KD cells (Supplementary Data 2) and data previously obtained for STAG2 KD cells[8] (Supplementary Data 3). Gene Set Enrichment Analysis (GSEA) with GSEA_4.2.3 software[62] was used to compare the gene expression changes of NIPBL KD and STAG2 KD cells with the gene sets for "CdLS upregulated genes" and "CdLS down-regulated genes" (Supplementary Fig. 7c) comprising the deregulated genes with FDR < 0.01 found in lymphoid cell lines from CdLS patients[40] (Supplementary Data 4).

### Reporting summary

Further information on research design is available in the Nature Portfolio Reporting Summary linked to this article.

## Data availability

The data that support this study are available from the corresponding author upon reasonable request. The ChIP-seq, RNA-seq and Hi-C data generated for this study have been deposited in GEO, accession number GSE207116. Additional datasets used include GSE101921 for ChIP-seq, RNA-seq and Hi-C data in STAG2 KD MCF10A cells, GSE98551 for CTCF distribution in MCF10A cells and GSE12408 for expression profiling arrays of CdLS probands. Uncropped immunoblots and RT-qPCR data are presented in the Source data file while files with flow cytometry and immunofluorescence data presented in this article are available at OneDrive using the link https://fundacioncnio-my.sharepoint.com/:f:/g/personal/alosada_cnio_es/Ela_6Yn_LqpGvwUk9UIx4FcBDnqN64CjNW7lqSflNqDi8g?e=6BsyyV. Source data are provided with this paper.

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

## Acknowledgements

We are grateful to R.G. Syljuåsen (Oslo U.H.) and Lola Martínez (Flow Cytometry Unit, CNIO) for advice on the flow cytometry protocol, Diego Megías (Confocal Microscopy Unit) for analysis of microscopy images, Álvaro Quevedo for his contribution to initial ChIP-seq and RNA-seq analyses and the rest of the members of the Chromosome Dynamics and DNA Replication groups at CNIO for helpful discussions. We also thank K. Shirahige (Tokyo University) for the ESCO1 antibody, J. Méndez (CNIO) for MCM3 and ORC2 antibodies, and E. de Alava (IBIS) for the A673 cell line. This work has been funded by grant PID2019-106499RB-I00 from Agencia Estatal de Investigación (AEI/10.13039/501100011033), Ministerio de Ciencia e Innovación, to A.L. D.A.G. is the recipient of FPI fellowship BES-2017-080051 and D.G.-L. is supported by a grant from the Spanish Association against Cancer (AECC).

## Author contributions

A.C. performed ChIP-seq and Hi-C experiments and generated A673 cell clones used in this study; D.G.-L. analyzed ChIP-seq, Hi-C and RNA-seq data; D.A.G. performed and analyzed all other experiments; M.R.-C. provided technical help for cloning, immunoblotting and custom antibody generation and characterization; A.L. supervised the study and wrote the manuscript with input from all authors.

## Competing interests

The authors declare no competing interests.
