## [Peer Review File · Nature Communications]

Different NIPBL requirements of cohesin-STAG1 and cohesin-STAG2Editorial Note: This manuscript has been previously reviewed at another journal that is not operating a transparent peer review scheme. This document only contains reviewer comments and rebuttal letters for versions considered at *Nature Communications* .

REVIEWER COMMENTS

Reviewer #3 (Remarks to the Author):

The revised manuscript by Alonso Gil et al has improved significantly. The authors have overall addressed the points that I raised during the initial review. Only a few small issues remain:

There are a couple of instances where the authors still refer to cohesin's chromatin association as 'loading' rather than as the more appropriate 'chromatin association'. These instances include the first paragraph of the results section, and the title of Figure 6. Please make sure that this phrasing is correct throughout the whole manuscript.

In the text it is made clear that STAG1 accumulates at CTCF sites in a setting when residual CTCF remains there despite its partial CTCF knockdown. It would be good to see this reflected also in the figures. I would recommend to incorporate into the supplemental figures that differential plot as included in the rebuttal to referees. This figure clearly shows that STAG1 is depleted at CTCF sites when no CTCF is present, which would be beneficial for all readers to see.

The role of NIPBL in differentiating STAG1-cohesin and STAG2-cohesin is novel and of interest to the field. With these suggested minor amends, the manuscript in my view would be ready for publication!

Reviewer #4 (Remarks to the Author):

In this manuscript by Gil et al., the authors use flow cytometry assay, chromatin fractionation followed immunoblotting, and calibrated ChIP-seq to dissect the role of NIPBL on STAG1 and STAG2 chromatin binding. The main point of the paper is that NIPBL knockdown appears to cause the STAG1 to accumulate at CTCF sites, while STAG2 decreases its binding genome-wide. The fact that NIPBL seems to play a central role in Cohesin loading with MAU2. These assays are a real strength of the paper. The questions I found myself asking while reading this paper are: 1) is this effect a direct effect on STAG1 or STAG2? 2) is there a convincing connection between STAG1 or STAG2 subunit and the entire Cohesin complex? I found the evidence not persuasive for both, and my overall impression is that the reach of this paper has exceeded its grasp.

1. The ChIP-seq assay is given a cursory explanation. I do not think the authors provide evidence for the increase of Cohesin-STAG1. The Cohesin (SMC1A) is actually decreased. Thus, it is difficult for me to understand the claim that NIPBL is not required for the association of Cohesin with chromatin. Moreover, they only examined the STAG1 instead of Cohesin-STAG1 variant complex. I would suggest either they perform reChIP analyses for SMC1A and STAG1 or adjust their statements.

2. This revision has mentioned that 10%-15% of NIPBL is left after RNAi, which may explain the increased loading of STAG1. Even though similar results were observed in different cell lines and different siRNAs, the changes of STAG1 are minor in their assay. Did the author also quantify the percentage of changes for the immunofluorescence or identify differential ChIP-seq binding peaks by using stringent cut-off and statistics? Considering the total levels of STAG1 and STAG2 proteins were changed, I do not believe the authors provided direct evidence for the NIPBL functions on STAG2 chromatin binding. Maybe the acute protein degradation technique could avoid the interference of the

secondary effects of gene expression due to the long-term perturbation of RNAi.

3. For the flow cytometry assay, most of the data did not provide quantitative and rigorous statistical analyses and raw images. Without this, it is difficult to interpret the effects of RNAi are significant or not.

4. Figure 4B. It is quite strange to me that the decrease of STAG2 is even more dramatic than the CTCF itself after CTCF RNAi.

5. For the Hi-C analyses, the author only performed the aggregate analyses. Do they also examine the 3D chromatin interactions changes for the regions with different magnitude (high/medium/low) changes of STAG1 or STAG2 chromatin binding, or regions with opposite effect or similar, or no effects on STAG1 and STAG2, to further explore the potential biological functions of NIPBL regulations on STAG1 or STAG2?

Reviewer #5 (Remarks to the Author):

In relation to Reviewer 1, authors have addressed each comment, supplying new experiments, including controls, making text modifications and providing convincing explanations. I think that overall the manuscript has improved. The provided experiments and controls continue in line with previous results. According to Reviewer's comments they have changed the title, which now better describes the main achievement of this work. It is clear that NIPBL knock down (KD) affects cohesin-STAG1 and cohesin-STAG2 very differently. However, it remains also clear that in the absence of a total loss of NIPBL it is hard to argue against NIPBL as a cohesin loader. It seems that this message has been moderated, stating in the Discussion that it cannot be excluded that the small amount of NIPBL left after KD is sufficient to support cohesin loading. However, in the Abstract and also in the Discussion, authors propose a model in which NIPBL is not required for the initial association of cohesin with chromatin. As formerly stated, and as reflected in the manuscript, some of the results can be at least partly explained by altered protein expression. It is obvious that NIPBL KD is having a clear impact on chromatin topology and transcription. To me, the relevance of this work relies in the differential behavior of cohesin-STAG1 and cohesin-STAG2 under low NIPBL levels, which is well documented. I find that data are not convincingly supporting that NIPBL is not involved in cohesin loading, and that this hypothesis must remain speculative.

In relation to Reviewer 2, authors have also addressed each comment. These comments mainly focus on conflicts of authors results with previous publications. I find that the main point of the work, i.e., the differences between cohesin-STAG1 and cohesin-STAG2 under low NIPBL levels are not in conflict with previous publications. Regarding conflicts raised by the Reviewer, authors give appropriate explanations for all of them, mostly based on higher performance of their analyses compared with former publications. Of note, the Reviewer highlights the FACS technique used as a novel and very interesting approach. Thus, FACS, calibrate ChIP-seq and additional techniques allow refinement of the analysis, leading to more precise observations that in some cases cannot be fully compared with previous approaches. In line with Reviewer 1, Reviewer 2 also claims that initial title is not supported by experiments presented and accordingly, authors have changed the title. However, as formerly indicated, despite the new title, in several parts of the manuscript the initial message still remains in a way that is not as speculative as would be appropriate.

Reviewer #3 (Remarks to the Author):

The revised manuscript by Alonso Gil et al has improved significantly. The authors have overall addressed the points that I raised during the initial review. Only a few small issues remain:

There are a couple of instances where the authors still refer to cohesin's chromatin association as "loading" rather than as the more appropriate "chromatin association" These instances include the first paragraph of the results section, and the title of Figure 6. Please make sure that this phrasing is correct throughout the whole manuscript.

We have replaced "loading" in the following instances:

- page 4, first paragraph of Results:

-NIPBL KD does not prevent association of cohesin with chromatin and has opposite effects on STAG1 and STAG2

-To assess the presence of cohesin on chromatin throughout the cell cycle in individual cells...

- page 5

Thus, the opposite effect of NIPBL KD in chromatin-bound levels of cohesin-STAG1 and cohesin-STAG2...

- page 7

...cohesin-STAG1 binds chromatin preferentially at or near CTCF positions.

- page 27

Figure 3. NIPBL KD increases cohesin levels on chromatin in STAG2 KO cells

- page 28.

Figure 6 An alternative model for the role of NIPBL in chromatin association and extrusion by the two cohesin variants.

In the text it is made clear that STAG1 accumulates at CTCF sites in a setting when residual CTCF remains there despite its partial CTCF knockdown. It would be good to see this reflected also in the figures. I would recommend to incorporate into the supplemental figures that differential plot as included in the rebuttal to referees. This figure clearly shows that STAG1 is depleted at CTCF sites when no CTCF is present, which would be beneficial for all readers to see.

We have now incorporated the differential plot mentioned by the reviewer to revised Figure S5.

The role of NIPBL in differentiating STAG1-cohesin and STAG2-cohesin is novel and of interest to the field. With these suggested minor amends, the manuscript in my view would be ready for publication!

Reviewer #4 (Remarks to the Author):

In this manuscript by Gil et al., the authors use flow cytometry assay, chromatin fractionation followed immunoblotting, and calibrated ChIP-seq to dissect the role of NIPBL on STAG1 and STAG2 chromatin binding. The main point of the paper is that NIPBL knockdown appears to cause the STAG1 to accumulate at CTCF sites, while STAG2 decreases its binding genome-wide. The fact that NIPBL seems to play a central role in Cohesin loading with MAU2. These assays are a real strength of the paper. The questions I found myself asking while reading this paper are: 1) is this effect a direct effect on STAG1 or STAG2? 2) is there a convincing connection between STAG1 or STAG2 subunit and the entire Cohesin complex? I

found the evidence not persuasive for both, and my overall impression is that the reach of this paper has exceeded its grasp.

1. The ChIP-seq assay is given a cursory explanation. I do not think the authors provide evidence for the increase of Cohesin-STAG1. The Cohesin (SMC1A) is actually decreased. Thus, it is difficult for me to understand the claim that NIPBL is not required for the association of Cohesin with chromatin. Moreover, they only examined the STAG1 instead of Cohesin-STAG1 variant complex. I would suggest either they perform reChIP analyses for SMC1A and STAG1 or adjust their statements.

The reviewer raises a valid point, which we also asked ourselves when we first saw the flow cytometry profile of STAG1 after NIPBL KD. To answer this question, we performed the experiment shown in Figure 1D, in which we co-deplete SMC1 and NIPBL. Under this condition, we fail to see an increase for STAG1 levels on chromatin, and thus we conclude that the increase that we observe in STAG1 occurs in the context of the whole cohesin-STAG1 complex. We have rewritten this part in main text to clarified this point in first section of Results. We reckon that the same is true for the ChIP-seq experiment. The decrease observed in SMC1 signals after NIPBL KD is the result of the strong decrease in STAG2, which is not compensated by the increase in STAG1 (see normalized read density plots in Figure 4C, left). Moreover, we previously showed that cohesin-STAG2 is around 1.5-2 fold more abundant than cohesin STAG1 in MCF10A cells used for the ChIP analyses (Kojic et al 2018 NSMB).

2. This revision has mentioned that 10%-15% of NIPBL is left after RNAi, which may explain the increased loading of STAG1. Even though similar results were observed in different cell lines and different siRNAs, the changes of STAG1 are minor in their assay. Did the author also quantify the percentage of changes for the immunofluorescence or identify differential ChIP-seq binding peaks by using stringent cut-off and statistics? Considering the total levels of STAG1 and STAG2 proteins were changed, I do not believe the authors provided direct evidence for the NIPBL functions on STAG2 chromatin binding. Maybe the acute protein degradation technique could avoid the interference of the secondary effects of gene expression due to the long-term perturbation of RNAi.

We agree with the reviewer that the changes are not as large as expected, and this is in fact the point of this manuscript: that with little NIPBL in the cell (10-15%), the effect on chromatin bound cohesin is rather small. If, as the reviewer suggests, the changes observed in chromatin bound STAG1 and STAG2 are the consequence of changes in STAG1 and STAG2 levels and not in NIPBL levels, this is also consistent with our proposal that the amount of NIPBL does not dictate chromatin binding. Thus, as we mention in the previous response to reviewers and in the Discussion: “While we cannot discard that the small amount of NIPBL left after KD may be sufficient to load all the cohesin that we detect bound to chromatin under this condition, an alternative possibility is that association of cohesin with chromatin does not require NIPBL (Fig. 6).”

The point of the Immunofluorescence experiment is to show that results are similar to the ones obtained with the flow cytometry assay.

Finally, we agree with the reviewer that acute degradation with a degron system, for instance, would be useful if NIPBL levels can be further reduced and also because acute degradation instead of the slower decrease resulting from siRNA may affect differently the transcriptional response of *STAG1*. So far, we have been unable to obtain such tool.

3. For the flow cytometry assay, most of the data did not provide quantitative and rigorous statistical analyses and raw images. Without this, it is difficult to interpret the effects of RNAi are significant or not.

The raw data for all the flow cytometry experiments can be provided upon request. As for quantitative analyses, we have now included a new panel in Supplementary Figure S2 in which we compare mean intensity values for STAG1 and STAG2 in NIPBL KD and control HeLa cells from four different flow cytometry experiments. The plot shows an increase in STAG1 signal and a decrease in STAG2 signal upon

NIPBL KD, both statistically significant (p-values: 0.02 and 0.008686, respectively, using paired Student t Test)

4. Figure 4B. It is quite strange to me that the decrease of STAG2 is even more dramatic than the CTCF itself after CTCF RNAi.

Our interpretation is that the CTCF-bound sites that remain after CTCF KD are occupied by STAG1. As per suggestion of the reviewer #3, we have now included a new panel in Supplementary Figure S5 (B), which we had prepared for our previous response to reviewers, which emphasizes this conclusion. In this panel (pasted below) heatmaps show the changes in CTCF, STAG1 and STAG2 accumulation at CTCF-cohesin sites in CTCF KD vs control cells. Positions in the upper half, in which CTCF decreases less, show increased STAG1. Positions in the lower half, in which the loss of CTCF signal is more pronounced, STAG1 also decreases. STAG2 decreases more in those positions in which STAG1 increases, suggesting competition of the two complexes.

5. For the Hi-C analyses, the author only performed the aggregate analyses. Do they also examine the 3D chromatin interactions changes for the regions with different magnitude (high/medium/low) changes of STAG1 or STAG2 chromatin binding, or regions with opposite effect or similar, or no effects on STAG1 and STAG2, to further explore the potential biological functions of NIPBL regulations on STAG1 or STAG2?

We performed some analyses asking about the changes in cohesin STAG1 and STAG2 in the anchors of loops that were gained, lost or shared between control and NIPBL KD cells (image pasted below). We observed that STAG2 was similarly lost in all loops. STAG1 was gained in all positions, but the increase was larger in gained loops, probably because these positions had lower STAG1 occupancy in control cells.

The behavior of SMC1 was the result of combining STAG1 and STAG2: it decreased at all positions, but less in those in which STAG1 increased more. The anchors of shared loops showed the highest occupancy for all three subunits in control cells. These analyses have been added to Supplementary Figure 6.

Reviewer #5 (Remarks to the Author):

In relation to Reviewer 1, authors have addressed each comment, supplying new experiments, including controls, making text modifications and providing convincing explanations. I think that overall the manuscript has improved. The provided experiments and controls continue in line with previous results. According to Reviewer's comments they have changed the title, which now better describes the main achievement of this work. It is clear that NIPBL knock down (KD) affects cohesin-STAG1 and cohesin-STAG2 very differently. However, it remains also clear that in the absence of a total loss of NIPBL it is hard to argue against NIPBL as a cohesin loader. It seems that this message has been moderated, stating in the Discussion that it cannot be excluded that the small amount of NIPBL left after KD is sufficient to support cohesin loading. However, in the Abstract and also in the Discussion, authors propose a model in which NIPBL is not required for the initial association of cohesin with chromatin. As formerly stated, and as reflected in the manuscript, some of the results can be at least partly explained by altered protein expression. It is obvious that NIPBL KD is having a clear impact on chromatin topology and transcription. To me, the relevance of this work relies in the differential behavior of cohesin-STAG1 and cohesin-STAG2 under low NIPBL levels, which is well documented. I find that data are not convincingly supporting that NIPBL is not involved in cohesin loading, and that this hypothesis must remain speculative.

In relation to Reviewer 2, authors have also addressed each comment. These comments mainly focus on conflicts of authors results with previous publications. I find that the main point of the work, i.e., the differences between cohesin-STAG1 and cohesin-STAG2 under low NIPBL levels are not in conflict with previous publications. Regarding conflicts raised by the Reviewer, authors give appropriate explanations for all of them, mostly based on higher performance of their analyses compared with former publications. Of note, the Reviewer highlights the FACS technique used as a novel and very interesting approach. Thus, FACS, calibrate ChIP-seq and additional techniques allow refinement of the analysis, leading to more precise observations that in some cases cannot be fully compared with previous approaches. In line with Reviewer 1, Reviewer 2 also claims that initial title is not supported by experiments presented and accordingly, authors have changed the title. However, as formerly indicated, despite the new title, in

several parts of the manuscript the initial message still remains in a way that is not as speculative as would be appropriate.

We thank this reviewer for revising carefully not only the manuscript but also the response to reviewers of the previous round of review. We have made additional changes to the manuscript to further tone down our claims, as requested:

- In abstract, instead of “our data support a model”, we say “Our data are consistent with a model in which NIPBL may not be required for chromatin association of cohesin...”

-By end of Introduction:

“we provide evidence of their different requirements for the putative loader and further speculate that initial association of cohesin with chromatin may be independent of NIPBL.”

-Also, in the model in Figure 6, instead of “current model” and “new model” we now say “model I” and “model II” and in the title we refer to an alternative model

Figure 6. An alternative model for the role of NIPBL in chromatin association and extrusion by the two cohesin variants.

In model I, NIPBL-MAU2 promotes chromatin association of cohesin (STAG1 or STAG2) as well as loop extrusion until the complex is released by WAPL or becomes arrested by CTCF proteins bound in convergent orientation. In the alternative model II, NIPBL-MAU2 is not required for association of cohesin with chromatin, but it is for loop extrusion.

-And in main text, Discussion, we have also rephrased the corresponding sentence:

While we cannot discard that the small amount of NIPBL left after KD may be sufficient to load all the cohesin that we detect bound to chromatin under this condition, an alternative possibility is that association of cohesin with chromatin does not require NIPBL (Fig. 6).

REVIEWERS' COMMENTS

Reviewer #4 (Remarks to the Author):

The authors have done some work related to my concerns, but my two major concerns were just ignored. Thus, it is still unclear whether it is a direct effect on STAG1 or STAG2. The work relies heavily on correlations or explanations, and the effect size is relatively small, which could be explained by an alternative hypothesis. Given these issues, I am not sure what the authors are observing and how this relates to Cohesin biology. Specific concerns are listed below.

1. For the Cohesin-STAG1 or Cohesin-STAG2 complex. The authors explained that co-depletion of SMC1 and NIPBL failed to see an increase of STAG1 in chromatin. The abundance of Cohesin-STAG1 and Cohesin-STAG2 are different etc. Their explanations cannot directly prove that the examination only STAG1 but stated Cohesin-STAG1 complex at all. My suggestions of reChIP or adjusting statements were just ignored.

2. The effects observed with NIPBL RNAi are fairly subtle. To acutely represent the effects, I suggested quantifying the percentage of fold changes for the immunofluorescence signals or identifying differential ChIP-seq binding peaks by using stringent cutoff and statistics. Without these numbers and statistics, it could not exclude the possibility the observed effects were experimental variations or not. Again, this suggestion was also ignored.

3. For the effects of CTCF RNAi, the authors explained the competition of STAG1 and STAG2. That is to say, the decrease in STAG2 was due to the secondary effects of CTCF RNAi (a slight increase of STAG1). This confirms my concerns about RNAi's indirect effects, and many other potential secondary effects were not investigated at all.

Reviewer #5 (Remarks to the Author):

I think the authors have met all my requirements and for me the manuscript is suitable for publication

Response to reviewers R3

Reviewer #4 (Remarks to the Author):

The authors have done some work related to my concerns, but my two major concerns were just ignored. Thus, it is still unclear whether it is a direct effect on STAG1 or STAG2. The work relies heavily on correlations or explanations, and the effect size is relatively small, which could be explained by an alternative hypothesis. Given these issues, I am not sure what the authors are observing and how this relates to Cohesin biology. Specific concerns are listed below.

1. For the Cohesin-STAG1 or Cohesin-STAG2 complex. The authors explained that co-depletion of SMC1 and NIPBL failed to see an increase of STAG1 in chromatin. The abundance of Cohesin-STAG1 and Cohesin-STAG2 are different etc. Their explanations cannot directly prove that the examination only STAG1 but stated Cohesin-STAG1 complex at all. My suggestions of reChIP or adjusting statements were just ignored.

Re-ChIP experiment are complex and, in this case, we find it unnecessary. As explained in our previous response, two pieces of evidence indicate that cohesin-STAG1 and not only STAG1 is bound to chromatin in the NIPBL KD condition:

- (1) Co-depletion of NIPBL KD and SMC1A KD reduces STAG1 levels on chromatin, as mentioned above by the reviewer, and
- (2) ChIP signals at CTCF sites in NIPBL KD condition increase for STAG1, decrease heavily for STAG2 and mildly for SMC1A. Changes in SMC1A ChIP signals are most likely the result from the combination of (increased) cohesin-STAG1 and (decreased) cohesin-STAG2.

Nevertheless, in the re-revised version (R3) of the manuscript, we have introduced the following changes:

- (1) we have toned down the conclusion from the first experiment:

Results, Page 4: “To exclude a cohesin-independent role of STAG1, we co-depleted SMC1A along with NIPBL. The increase of STAG1 on chromatin was abrogated under this condition, *suggesting* that it occurs in the context of a full cohesin complex (Fig. 1d).”

- (2) we discuss as potential caveat the reviewer’s point:

Discussion, Page 10: “The decrease in SMC1A accumulation at CTCF sites observed in NIPBL KD cells by ChIP-seq, much milder than the decrease in STAG2, further *suggests* that the whole cohesin-STAG1 complex, and not only STAG1, is present at these sites under this condition, *although re-ChIP experiments would be required to confirm this point* (Fig. 4c).

2. The effects observed with NIPBL RNAi are fairly subtle. To acutely represent the effects, I suggested quantifying the percentage of fold changes for the immunofluorescence signals or identifying differential ChIP-seq binding peaks by using stringent cutoff and statistics. Without these numbers and statistics, it could not exclude the possibility the observed effects were experimental variations or not. Again, this suggestion was also ignored.

The changes that we show are statistically significant:

1. For flow cytometry, we had already included in the previous version (R2) a graph showing that both the increase in STAG1 and the decrease in STAG2 after NIPBL KD are statistically significant (p-values: 0.02 and 0.008686, respectively, using paired two-sided Student t Test).

2. For the immunofluorescence experiment in Fig. 1c, we now include a new Supplementary Table 1 in which we calculate the percentage of change in mean intensity for STAG1, STAG2 and SMC1A between the control and NIPBL KD cells and show statistical significance (p values $<2e-16$) using a non-parametric Mann Whitney two-sided test with confidence intervals of 99%.

staining	condition	Mean intensity	SD	n° cells	% change	p value
SMC1	control	103.93	15.06	372	83%	$<2E-16$
	NIPBL KD	85.86	18.76	378		
STAG1	control	62.36	15.64	419	129%	$<2E-16$
	NIPBL KD	80.34	25.19	559		
STAG2	control	65.80	17.66	419	77%	$<2E-16$
	NIPBL KD	51.10	12.04	559		

3. For ChIP, we include now a new Supplementary Table 3 with the analysis of differential peaks in between control and NIPBL KD for each antibody with two cut-offs: $FDR < 0.05$ and more stringent $FDR < 0.01$, as requested. The results are quite similar for both cut-offs.

antibody_condition	FDR < 0.05			FDR < 0.01		
	peak number	common peaks	% peaks KD vs control	peak number	common peaks	% peaks KD vs control
SMC1A_control	35,171	12,742	45.3%	30,305	9,100	34.9%
SMC1A_NIPBL KD	15,945			10,574		
STAG1_control	26,355	22,524	110.9%	22,849	19,872	110.1%
STAG1_NIPBL KD	29,232			25,167		
STAG2_control	55,059	7,852	15.3%	46,577	5,936	13.4%
STAG2_NIPBL KD	8,413			6,247		

3. For the effects of CTCF RNAi, the authors explained the competition of STAG1 and STAG2. That is to say, the decrease in STAG2 was due to the secondary effects of CTCF RNAi (a slight increase of STAG1). This confirms my concerns about RNAi's indirect effects, and many other potential secondary effects were not investigated at all.

We apologize if we have not explained ourselves with sufficient clarity. When CTCF is not bound to chromatin, cohesin cannot arrest at CTCF sites and ChIPseq signal for cohesin should decrease at these sites. In our CTCF KD condition, ChIP-seq analyses indicate that there is CTCF remaining at many sites and the surprising observations are (1) that the decrease is not the same for STAG1 and STAG2, but much more dramatic for STAG2; and (2) that cohesin-STAG1 occupancy even increases at a fraction of sites (those in the upper part of heatmap in Supplementary Fig. 5b). The increase in *STAG1* mRNA levels most likely explains this last observation. Nevertheless, the important point is that the results of the CTCF KD experiments are consistent with the proposal that cohesin-STAG1 arrives first at CTCF sites, before cohesin-STAG2. We have rewritten the corresponding sentence in Discussion to clarify this:

Page 11, “Faster arrival of cohesin-STAG1 complexes to CTCF sites is also consistent with the prevalence of STAG1 over STAG2 at those CTCF-bound sites remaining after CTCF KD (Fig. 4b and Supplementary Fig. 5).

Reviewer #5 (Remarks to the Author):

I think the authors have met all my requirements and for me the manuscript is suitable for publication